# Investigating Urban Flooding and Nutrient Export under Different Urban Development Scenarios in the Rouge River Watershed in Michigan, USA

Yilun Zhao [1],*, Yan Rong [1], Yiyi Liu [1], Tianshu Lin [1], Liangji Kong [2], Qinqin Dai [3] and Runzi Wang [1]

1   School for Environment and Sustainability, University of Michigan, 440 Church Street,
Ann Arbor, MI 48109, USA; yanrong@umich.edu (Y.R.); yiyiliu@umich.edu (Y.L.); tianshul@umich.edu (T.L.);
runziw@umich.edu (R.W.)
2   Department of Real Estate, Cornell SC Johnson College of Business, Cornell University,
Ithaca, NY 14850, USA; lk547@cornell.edu
3   Leonard N. Stern School of Business, New York University, New York, NY 10012, USA; qd2120@stern.nyu.edu
*   Correspondence: yilunzh@umich.edu; Tel.: +1-734-277-8376

**Abstract:** Adverse environmental impacts in the watershed are driven by urbanization, which is reflected by land use and land cover (LULC) transitions, such as increased impervious surfaces, industrial land expansion, and green space reduction. Some adverse impacts on the water environment include urban flooding and water quality degradation. Our study area, the Rouge River Watershed, has been susceptible to accelerated urbanization and degradation of ecosystems. Employing the Land Change Modeler (LCM), we designed four alternative urban development scenarios for 2023. Subsequently, leveraging the Integrated Valuation of Ecosystem Services and Tradeoffs (InVEST), we utilized two models—Nutrient Delivery Ratio (NDR) and Flood Risk Mitigation (UFRM)—to evaluate and compare the performance of these scenarios, as well as the situation in 2019, in terms of nutrient export and urban flooding. After simulating these scenarios, we determined that prioritizing the medium- and high-intensity development scenario to protect open space outperforms other scenarios in nutrient export. However, the four scenarios could not exhibit significant differences in urban flooding mitigation. Thus, we propose balanced and integrative strategies, such as planning green infrastructure and compact development, to foster ecological and economic growth, and enhance the Rouge River Watershed's resilience against natural disasters for a sustainable future.

**Keywords:** urban flooding; nutrient export; LULC; sustainable development; Great Lakes region

## 1. Introduction

Land use and land cover (LULC) change and land management decisions in urban regions critically modulate flooding and nutrient export in water systems, altering the availability and quality of water resources [1]. The correlation between land and water management is affected by multiple factors, including climate change, soil hydrological conditions, and shifts in urban political and economic landscapes [2]. Excess nitrogen (N) and phosphorus (P) exports can lead to environmental, human health, and economic impacts through the eutrophication of water bodies. For instance, harmful algal blooms (HABs) in Lake Erie have caused severe damage to the local ecosystem and economy due to water eutrophication [3]. Due to urban flooding, increased surface runoff from impervious surface expansion can lead to urban drainage and deluge [4,5]. In addition, urban flooding can also impact nutrient export by modifying nutrient concentrations in the high-flow event [4–7]. Therefore, comprehending the dynamics of N and P export and urban flooding is essential for regional watershed management. In this study, we utilized two models from the Integrated Valuation of Ecosystem Services and Tradeoffs (InVEST): Nutrient Delivery Ratio (NDR) and Urban Flooding Risk Mitigation (UFRM) models. The NDR model is

known for its computational stability and ability to estimate water quality parameters and simulate nutrient transport over extended periods [8]. On the other hand, the UFRM model can compute surface runoff and evaluate the corresponding flood risk value through the straightforward Soil Conservation Service Curve Number (SCS-CN) method [9].

Protecting open space and enhancing the intensity of existing urban regions is a rational and sustainable pattern for urban development. The rich functionality of urban open space creates room for human activities and acts as green space to modulate regional ecology by reducing surface runoff, improving air quality, mitigating soil pollution, and providing different characteristic ecosystem services [10–12]. However, growing human interventions due to urbanization have transformed these open spaces into fragmented patches [13,14]. Consequently, these alterations can diminish their overall accessibility and significantly reduce their original ecological and social functions [15]. In addition to open spaces, high-intensity development as an urban planning strategy can synchronize human requirements with land optimization. It enhances ecosystem services and supports the efficacy of economic growth and infrastructure development [16]. Moreover, research studies have highlighted other significant benefits of high-intensity development patterns, including shorter travel distances and lower individual energy consumption [17–19]. This development model can also optimize resource utilization in infrastructure projects and preserve green open space and natural regions by minimizing development on other LULC types [20].

LULC is impacted by the combination of natural factors and human interventions, primarily attributed to climate change, soil conditions, population density, infrastructure development, urban politics, and the economy [2]. This interaction introduces several uncertainties and complexities to urban planning at varying spatial and temporal scales [21]. In this study, we utilized the Land Change Modeler (LCM) to predict future land cover changes. This modeler is advantageous in its dynamic projection capability and spatial simulation accuracy in urban growth and environmental conservation to construct prospective LULC scenarios for 2030, which are closely related to urban development intensity and ecological sustainability [22]. However, most LCM-based studies focus on macro changes in LULC classifications, which are often limited to a few types like cities, forests, farmlands, water, and wetlands. Particularly for urban LULC types, there is a lack of more detailed downward classifications. Furthermore, many studies focus on green spaces, especially forest transitions, with few exploring urban-related land type change analysis and simulation and creating related LULC scenarios.

The Rouge River in Southeastern Michigan has a long history of industrialization and consequent ecological degradation, persistent urban flooding, and water contamination [23,24]. Past research studies have illuminated the environmental challenges within the watershed from the biological and chemical perspectives [25,26]. Nonetheless, the literature gap persists in employing quantitative methodologies to predict and visualize the watershed's future urban development models and formulate sustainable strategies to address pressing water issues. Furthermore, based on our past analyses, variations in LULC are likely to impact urban flooding and nutrient export. Thus, investigating the spatiotemporal changes in LULC is beneficial for mitigating urban flooding and minimizing nutrient export in the Rouge River Watershed. Moreover, it can help in the sustainable governance of the watershed's future landscape and ecosystem. Given the above research gaps, this study aims to specify LULC types that can effectively mitigate the Rouge River Watershed's flooding and nutrient export issues for sustainable urban development. The primary objectives of our research are: (a) to quantify historical (2019) urban flooding and nutrient export of the Rouge River Watershed; (b) to evaluate the past LULC transformations in the watershed and explore future LULC scenarios based on open space preservation and compact development priority; and (c) to investigate the effects of LULC change on flooding and nutrient export, as well as the appropriate planning and urban development strategies for protecting the water environment.

## 2. Materials and Methods

### 2.1. Study Area and Its Environmental Challenges

The study area—Rouge River Watershed (83°6′–83°39 W, 42°13′–42°38 N), in Southeast Michigan—drains 467 square miles (1209.52 km$^2$) into the Detroit River (Figure 1). It comprises four major branches with 127 river miles (204.39 km) and numerous tributaries. Over 1.35 million people in 47 municipalities live within the watershed, including Detroit, the largest city in Michigan. In the past decade or so, the Rouge River Watershed has experienced rapid urbanization. From 2001 to 2019 developed medium-intensity areas underwent an expansion of 60.50%, and high-intensity developed areas increased by 38.90%. The land of Rouge River Watershed is more than 50% urbanized, with less than 25% still undeveloped.

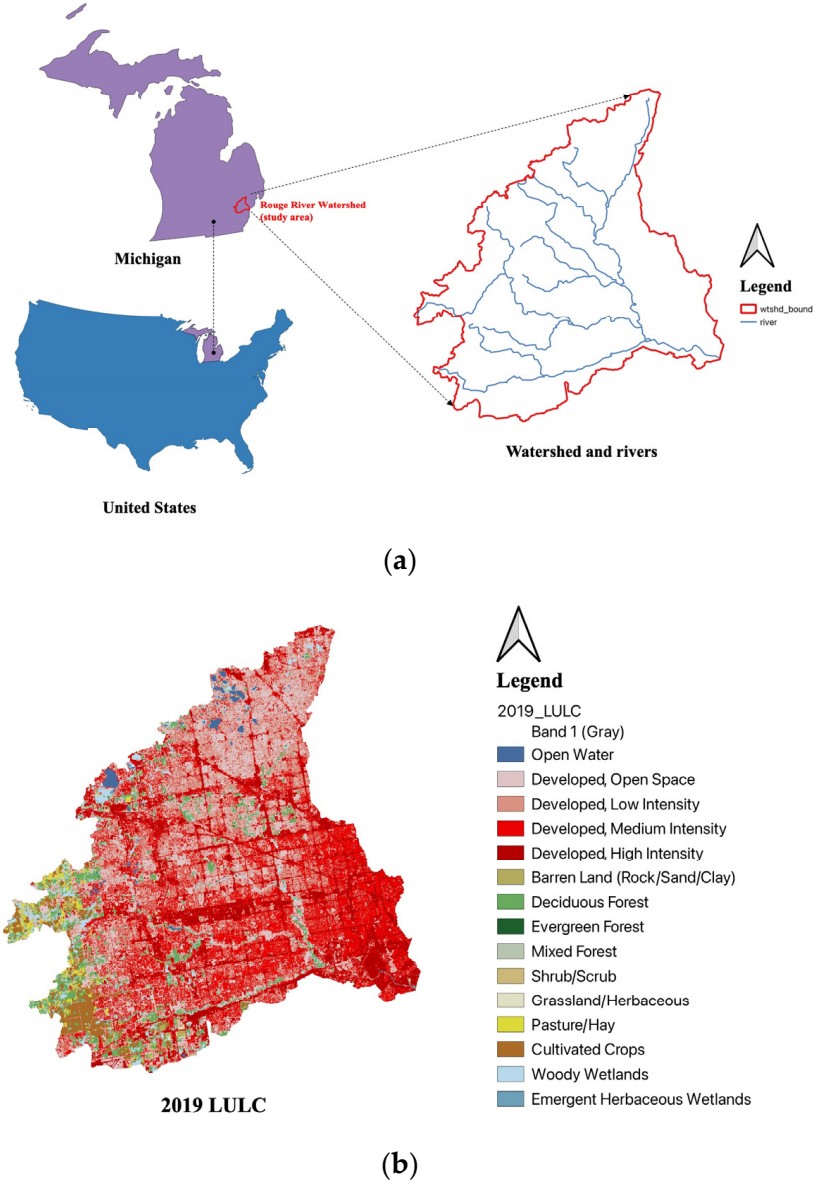

**Figure 1.** Study area and subregions. In (**a**), the bottom left map illustrates the location of Michigan State within the United States. The top left map depicts the Rouge River Watershed in Michigan State. The top right map delineates the Rouge River Watershed and its primary rivers. (**b**) Showcases the land use and land cover (LULC) within this watershed in the 2019 National Land Cover Database (NLCD).

While the Rouge River Watershed is vital in sustaining regional ecosystems, maintaining regional hydrology, and meeting diverse societal water requirements, its function is jeopardized by accelerating urbanization. Heavy industry pollutants and changes in LULC have degraded water quality and aquatic ecosystems [27]. Research in 2021 reported that 25 compounds were found at Rouge River mouth. By comparing with the other five sampling points, the number of detected compounds was highest downstream in the Detroit River found at Rouge River mouth. In addition, for the cumulative concentration of synthetic sweeteners concentration, Rouge River mouth is 60 times higher than Clinton River mouth, 8 times higher than Lake St. Clair Metropark, and 1600 times higher than Northeast Belle Isle and Southwest Belle Isle. The various types of detected contaminants (such as naproxen, carbamazepine, etc.), and higher concentrations of contaminants in the area indicate that the downstream Lake Huron to Erie corridor (including Rouge River watershed) is dominated by wastewater effluent and under severe pollution threat. The threat may be related to urbanization and developed industry [28]. The accelerating urbanization also threatened the watershed by various other pollution sources. For instance, aging infrastructure introduces combined sewer overflows (CSOs) and sanitary sewer overflows (SSOs). In addition, domestic waste and fertilizers introduce non-point source pollution. Moreover, the growth in impervious surface area disrupts natural hydrological processes and increases the susceptibility to urban flooding [29–33]. Given the extensive manufacturing, food, and other industries around the Rouge River and their history of discharging pollutants into the river [34], as well as the increased impermeable surfaces due to rapid urbanization, this watershed may face increasingly severe ecological threats (e.g., pollution, deforestation, etc.), in the future.

### 2.2. Nutrient Delivery Model (NDR) and Urban Flood Risk Mitigation Model (UFRM)

#### 2.2.1. NDR Model Foundation

The Nutrient Delivery Ratio (NDR) model of InVEST (version 3.13.0) is designed to map the sources of N and P, as well as to depict the relative nutrient export and retention within the watershed [8,35]. This model employs a mass balance approach, computing the nutrient loads of the study area based on nutrient loading rates for various LULC types [6]. The model formulates the nutrient output for each pixel i as the product of the load and the NDR, which is composed of both surface and subsurface components (1):

$$x_{exp,i} = load_{surf,i} * NDR_{surf,i} + load_{subs,i} * NDR_{subs,i} \tag{1}$$

In Equation (1), $load_{surf,i}$ and $load_{subs,i}$ specify the nutrient loads for surface and subsurface, respectively. Moreover, $NDR_{surf,i}$ and $NDR_{subs,i}$ denote the NDR for surface and subsurface, respectively. Detailed calculations for these parameters can be referred to in the "InVEST User's Guide" [6]. The final nutrient output is the sum of the contributions from all pixels within the watershed (2):

$$x_{exptotal} = \sum_i x_{exp,i} \tag{2}$$

#### 2.2.2. NDR Setup and Calibration

In executing the NDR model, three raster layers are required, including a digital elevation model (DEM) from NASADEM at 30 m resolution, a 2019 LULC map from the National Land Cover Database (NLCD), a runoff potential map from the Global Precipitation Measurement (GPM, version 6) at 30 m resolution, which was then converted into annual precipitation. Among these inputs, the NLCD utilizes spectral, spatial, temporal, and ancillary data as inputs and employs decision tree for classification [36,37]. For our study area, its grid cell size is 30 m by 30 m. In addition, the model also requires our area of interest (AOI), which is the boundary of the Rouge River Watershed from the Watershed Boundary Dataset (WBD) and a biophysical table. This table determines N and P loadings for diverse LULC types drawing from the literature [6,38]. Additionally, we set the subsur-

face critical length to 30 m, which is equal to the grid cell size [6]. We conducted a sensitivity test on the remaining three parameters: subsurface maximum retention efficiency (SMRF), threshold flow accumulation (TFA), and Borselli K parameter (kb). Among these parameters, SMRF is the maximum nitrogen retention efficiency that can be reached through subsurface flow. TFA is used to classify streams from the DEM. kb is a calibration parameter that determines the shape of the relationship between hydrologic connectivity and the nutrient delivery ratio [6]. The results (Figure A1) show that SMRF is not sensitive to the TN results, while TFA and kb are sensitive. Therefore, based on empirical formulas, we set the SMRF to 0.9 and calibrated TFA and kb [6]. We calibrated these two parameters through the correlation between nutrient export and the Stream Quality Index (SQI) for the Rouge River Watershed. The SQI is a water quality dataset formed by watershed environmental organizations by recording the population of macroinvertebrates at 113 sampling points within the watershed. The rationale of calibration is that SQI signifies sensitive organism abundance, which is impacted by nutrient export and the consequent eutrophication. If the model simulates nutrient export accurately, there will be a high correlation between nutrient export and SQI due to the agreement in their spatial pattern. Experimenting with varying TFA and kb values achieved the optimal correlation with TFA = 1000 and kb = 1, serving as the input of NDR modeling (complete calibration results are shown in Table A1).

### 2.2.3. UFRM Model Foundations

Employing the Urban Flood Risk Mitigation (UFRM) model from InVEST (version 3.13.0), we evaluated the urban flooding within our study area. The model quantified runoff reduction and the volume of retained runoff per pixel relative to the storm's volume. In this study, our primary output is Q, which denotes the runoff values measured in mm. Q is estimated using the curve number method, as formulated in Equation (1). The curve number (CN) indicates the runoff potential of the land, affected by various factors such as soil antecedent moisture condition, soil type, and land use. Its value ranges between 0 and 100 [9]. During the calculation process, the curve number (CN) is extracted from the Soil Hydrologic Group and Biophysical Table by the model [6]. We obtained the Soil Hydrologic Group from the gNATSGO database on the U.S. Department of Agriculture website. And for each pixel (i), defined by a land use type and soil attributes, the model estimates runoff (mm) with the CN method:

$$Q_{p,i} = \left\{ \begin{array}{l} \frac{\left(P - \lambda S_{max_i}\right)^2}{P + (1-\lambda) S_{max,i}} \, if \ P > \lambda \cdot S_{max,i} \\ 0 \ otherwise \end{array} \right\} \tag{3}$$

where P is the design storm depth in mm, $S_{max,i}$ is the potential retention in mm and $\lambda \, S_{max,i}$ is the rainfall depth required to initiate runoff, and it is also called the initial abstraction ($\lambda$ = 0.2 for simplification), where $S_{max}$ (calculated in mm) is a function of the curve number, CN, an empirical parameter that depends on land use and soil characteristics as Equation (4) [6]. The model then calculates runoff retention per pixel $R_i$ as Equation (5) and runoff retention volume per pixel $R\_m3_i$ as Equation (6). Finally, runoff volume per pixel $Q\_m3_i$ is computed using Equation (7) [35].

$$S_{max,i} = \frac{25400}{CNi} - 254 \tag{4}$$

$$R_i = 1 - \frac{Q_{p,i}}{P} \tag{5}$$

$$R\_m3_i = R_i \cdot P \cdot pixel.area \cdot 10^{-3} \tag{6}$$

$$Q\_m3_i = Q_{p,i} \cdot pixel.area \cdot 10^{-3} \tag{7}$$

2.2.4. UFRM Setup

Implementing the UFRM model requires AOI, rainfall depth, LULC, Soil hydrologic group, biophysical table, built infrastructure, and damage loss table. For LULC, we incorporated the same 2019 NLCD map utilized in the NDR model. Our AOI is the Rouge River Watershed. Furthermore, we adopted a 100-year 24h storm depth of 5.15 inches (130.81 mm) to represent rainfall depth in our study zone. We integrated a map of soil hydrologic groups from the Michigan gSSURGO data, processed and valued following distinct soil hydrologic groups. The map of building footprints, essential as built infrastructure, was obtained from the 2020 building footprint shapefile from the Southeast Michigan Council of Governments (SEMCOG) database. Damage loss table presents potential damage loss data for various building types. Since we did not need to calculate damage loss data for our study, we omitted it in our model [6].

*2.3. Scenario Design and Land Change Modeler (LCM)*

We implemented the Land Change Modeler (LCM) to generate future urban flooding and nutrient export evaluation scenarios. LCM uses the Cellular Automata (CA)_Markov model and transition susceptibility maps to assess and predict LULC change [39,40]. CA is a spatiotemporal dynamic model that interprets macro-level phenomena through interactions at the micro-level. These macro-level phenomena can vary widely including landscape change, socio-economic development, population migration, and biological pattern formation [41–43]. The Markov chain detects and forecasts changes in LULC [44,45]. Integrating the strengths of both CA and the Markov chain, the LCM can evaluate past LULC maps' temporal and spatial transformations to predict future LULC [46,47].

In our study, NLCD maps from 2001, 2011, and 2019 served as LULC data sources. In addition, we employed the LCM to analyze the historical trends in LULC changes and projected the potential LULC for 2030 based on its transition from 2011 to 2019. Within the LCM, we employed a multi-layer perceptron (MLP) neural network approach to evaluate the transfer potential of various LULC classes [48–50]. Analyzing past LULC changes, we retained the nine most significant LULC transition types (as presented in Table A1) as our transition sub-models. We designated the evidence likelihood of changes for all LULC classes and DEM as the driving variables for these sub-models. Among them, the evidence likelihood of changes is generated based on algorithm that comes with the LCM, while the DEM is the same input in the two InVEST models. Subsequently, using the transfer potential determined from the sub-model outputs within LCM, we predicted the 2030 future LULC. The forecasted LULC was set as our study's baseline scenario (S1).

To protect the watershed forest, we constrained the transition type from deciduous forest to open space in our sub-models and introduced a transition type from open space to deciduous forest. We only changed open space because other transitions from developed areas to forests are insignificant. Under this condition, we generated a forest conservation scenario (S2) for 2030. Given the historical LULC trends, it is evident that transitions between various types of urban LULC in this watershed have been dominant. Based on S2, we designed two additional scenarios: low-intensity development (S3) and medium- and high-intensity development (S4). S3 primarily constrained transitions from low- and medium-intensity developed areas to high-intensity areas, thus limiting the high-density regions in the watershed. Similarly, S4 restricted the transition from open space to low- and medium-intensity developed areas to determine the overdevelopment of open spaces, encouraging compact development on already developed LULC types. Table 1 presents the detailed descriptions and settings for the scenarios. Finally, we evaluated all scenario-based LULC maps using the NDR and UFRM models to choose the most beneficial urban development pattern for a healthy water ecosystem in the future.

**Table 1.** Settings and descriptions for the four LULC scenarios.

| Scenario Name | Limit Transition Type | Force Transition Type | Scenario Description |
|---|---|---|---|
| S1: Baseline | / | / | Keep original (2011–2019) 9 LULC's changes. |
| S2: Forest Conservation | Deciduous forest to developed, open space | Developed, open space to deciduous forest | Protect greenfield land types dominated by deciduous trees and convert open space to woodland. |
| S3: Low-Intensity Development | Deciduous forest to developed, open space Developed, low intensity to developed, high intensity Developed, medium intensity to developed, high intensity | Developed, open space to deciduous forest | Based on S1, limit the conversion of low and medium densities to high densities to minimize the creation of excessive high-intensity sites in urban development. |
| S4: Medium- and High-Intensity Development | Deciduous forest to developed, open space Developed, open space to developed, low intensity Developed, open space to developed, medium intensity | Developed, open space to deciduous forest | Based on S1, further, protect open space. Allow the city to develop incrementally on the original developed sites. |

## 3. Results

### 3.1. Historical LULC Changes

Based on the LULC of 2001, 2011, and 2019, we computed the net changes in LULC for three timespans—2001–2011, 2011–2019, and 2001–2019—to explore historical quantitative transformation patterns. We observed that from 2001 to 2019, developed medium-intensity areas underwent an expansion of 60.50%, and high-intensity developed areas increased by 38.90%, contrasted with reductions in developed open space and low-intensity developed regions by 41.65% and 27.04%, respectively (Figure 2). Simultaneously, moderate declines were observed in deciduous forests and cultivated crops by 14.45% and 10.66%. The changing patterns from 2011 to 2019 were similar to those from 2001 to 2019, with the rise of developed medium-intensity and high-intensity land and the cutback of developed open space and low-intensity land conversely. The net change patterns from 2000 to 2011 differ slightly from the other two time spans. Though developed medium-intensity and high-intensity land increased like the above two mentioned timespans, developed low-intensity areas were added instead of reduced. Significant declines occurred in the deciduous forests and cultivated crops rather than developed open spaces and low-intensity developed areas. Comparing the historical LULC change pattern (Figure A2), extensive modifications were explicitly evident in the Rouge River Watershed's northern and southwestern regions (Figure A2) from 2011 to 2019, revealing increased developed land and reduced cultivated cropland. Therefore, given the substantial changes during the 2011–2019 timespan, we selected the LULC data from 2011–2019 as our baseline for LCM simulations' four scenarios.

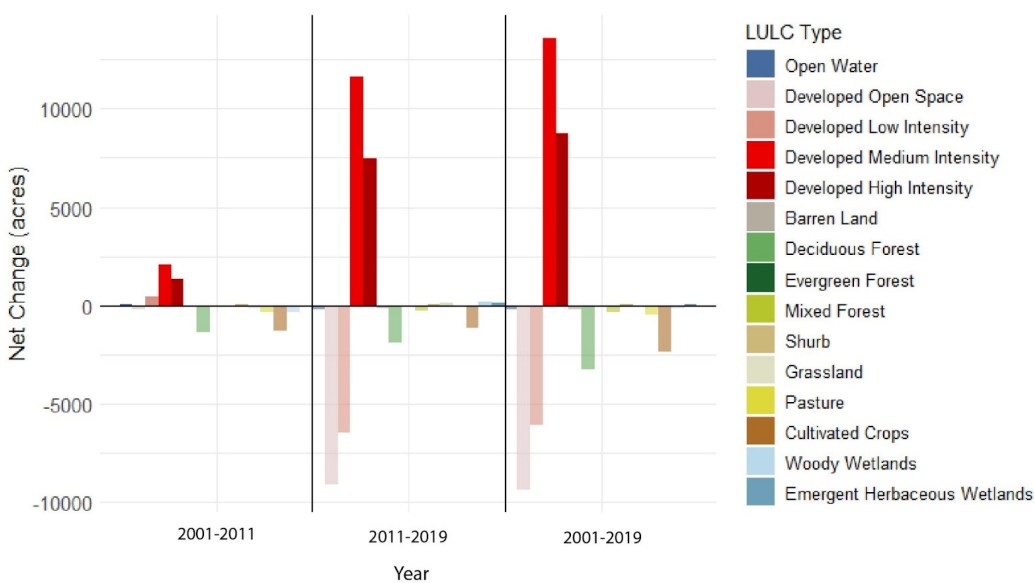

**Figure 2.** Bar chart of historical LULC net change in Rouge River Watershed. The diagram sequentially displays the net change in LULC within the Rouge River Watershed for three periods: 2001–2011 (left), 2011–2019 (middle), and 2001–2019 (right).

## 3.2. Variations in LULC under Future Scenarios

We evaluated considerable differences between the baseline and three alternative LCM scenarios (Figures A2 and 3). Under S1, where all LULC changes were allowed, open space experienced a significant reduction of 13,180 acres (−34.44%) (53.33 km$^2$), while high-intensity developed areas increased by 8114 acres (−17.80%) (32.84 km$^2$). Likewise, both medium- and low-intensity developed areas increased by 4520 (+5.14%) (18.29 km$^2$) and 1746 acres (+2.61%) (7.07 km$^2$), respectively.

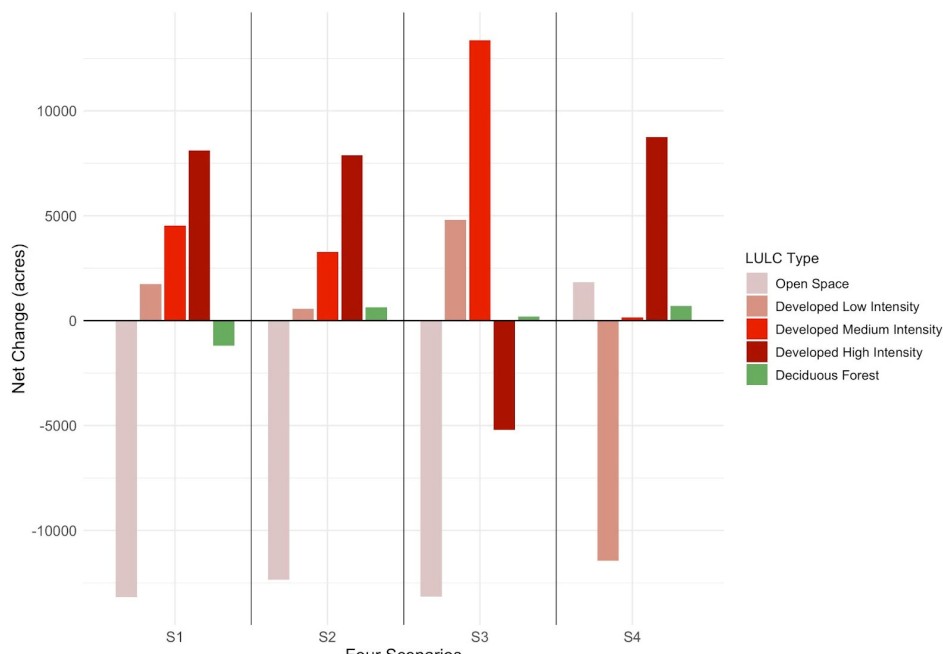

**Figure 3.** Bar chart of future scenarios' LULC net change in Rouge River Watershed. The diagrams sequentially present, from left to right, the net changes in five categories of (LULC) for four scenarios (S1, S2, S3, and S4) of the Rouge River Watershed in 2030. Five LULC categories include open space, low-intensity, medium-intensity, high-intensity, and deciduous forests.

Under S2, the forest conservation scenario, which restricts the transition from deciduous forest to open space, the latter reduced by 12,344 acres ($-31.57\%$) ($49.95$ km$^2$), while other land uses all followed an increasing trend. For instance, high-intensity developed areas increased by 7887 acres ($+17.38\%$) ($31.92$ km$^2$), and the deciduous forest area increased by 634 acres ($+3.72\%$) ($2.57$ km$^2$). S3 aims for a low-intensity development outcome, prohibiting high-intensity development based on S2. Under S3, open space and high-intensity developed areas decreased by $34.34\%$ and $16.14\%$, respectively. Likewise, medium-intensity developed areas and deciduous forests both increased, with the former exhibiting the most significant growth of 13,371 acres ($+13.81\%$) ($54.11$ km$^2$). The final scenario, S4, targets a medium- and high-intensity development outcome and preserves open space. Consequently, it results in a $21.30\%$ reduction in low-intensity developed areas, while the high-intensity developed areas conspicuously increased by 8747 acres ($+18.92\%$) ($3.54$ km$^2$).

Generally, open space has a decreasing trend in S1, S2, and S3, while it has an increasing trend in S4. Conversely, low-intensity developed areas increased in S1, S2, and S3, whereas they decreased in S4. Medium- and high-intensity regions generally increased in these scenarios, while the only exception was a decrease in S3. Deciduous forest exhibited a decline in S1 but growth in the other scenarios, most notably in S4 with a 699 acre ($2.83$ km$^2$) increase.

### 3.3. NDR Results

Export of N and P demonstrates spatial coherence upon NDR model evaluation of the Rouge River Watershed in 2019 (Figure A4). Both N and P showed massive exports in the southeastern, southern, central-southern, and some regions in the northwestern watershed. However, there was less N and P export in the northern and western regions. Subsequent revaluation of the four scenarios through NDR revealed consistent spatial patterns in N and P export.

The overall nutrient exports for N and P in the watershed for 2019 were 80,435 kg and 8462 kg, respectively (Table 2). The results confirm that all four scenarios exhibited significant growth compared to 2019. In particular, S2 experienced the most substantial increase in the export of N and P, at $8.47\%$ and $9.42\%$, respectively, while S4 displayed minimal growth with $5.18\%$ and $6.05\%$. Furthermore, S1 and S2 were similar in N and P increasing levels, at $8.37\%$ and $8.86\%$, and S3 and S4 were similar, at $5.34\%$ and $7.66\%$. Comparing the other three scenarios with the baseline (S1), we observed that the export of N and P in S2 increased by $0.10\%$ and $0.51\%$, while S4 demonstrated the most substantial reduction, at $2.94\%$ and $2.58\%$. In addition, the outcomes showed that the total export of N in the watershed was much higher than that of P in all scenarios, as well as in 2019.

**Table 2.** Total N and P contributions in the Rouge River Watershed for 2019 and four scenarios.

| Nutrient Export (kg) | 2019 | S1 | S2 | S3 | S4 |
|---|---|---|---|---|---|
| N | 80,435 | 87,165 (8.37%) [1] | 87,250 (8.47%) | 84,731 (5.34%) | 84,600 (5.18%) |
| P | 8462 | 9212 (8.86%) | 9259 (9.42%) | 9110 (7.66%) | 8974 (6.05%) |

[1] Percent change from N or P in 2019.

Regarding the spatial variations of the four scenarios compared to 2019, S1 experienced a significant elevation in N and P exports. Moreover, S2 presented a distinct rise in the north-central areas, while S3 demonstrated a generalized increase across the central regions and a decline in the northeastern areas. Contradictory to S3, S4 did not exhibit many fluctuations in N and P levels across large areas (Figure A4). It is obvious that the areas experiencing an upsurge in N and P in S4 were confined to a small region in the northeastern part of the watershed, but it was less pronounced than S1, with no significant changes observed in the other areas of the watershed (Figure 4).

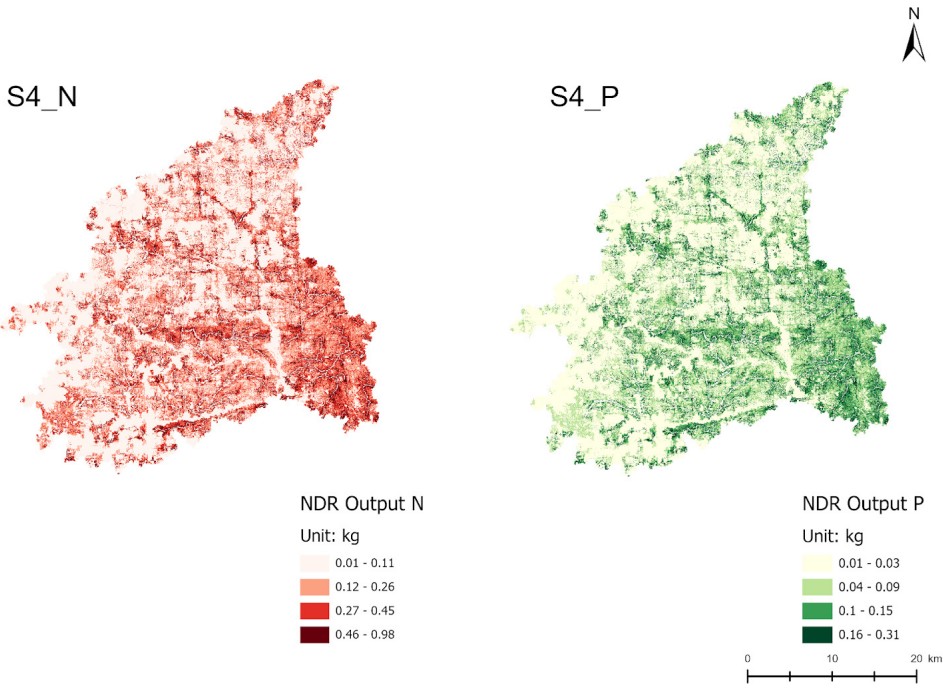

**Figure 4.** Map of the NDR output for S4. The map displays the N and P exports of the Rouge River Watershed for S4 which had the best performance in reducing nutrient exports among four scenarios.

*3.4. UFRM Results*

The four scenario simulations of UFRM are very similar to the 2019 UFRM outcomes for the Rouge River Watershed in both the runoff volume and spatial pattern. Through looking at the pattern of S3 (Figure 5), the surface runoff in the southern part of the site was generally more significant than that in the northern part of the site, and the surface runoff in the eastern part of the site was slightly larger than that in the western region. Moreover, some areas had obviously smaller surface runoff in the central site area, while the areas with more extensive surface runoff were mainly concentrated in the southeast corner of the site, as well as some areas in the middle and on the western boundary.

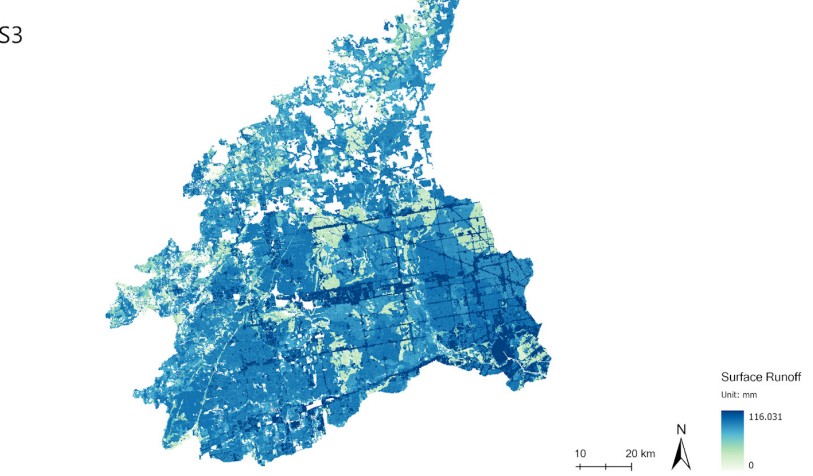

**Figure 5.** Map of the surface runoff of Rouge River Watershed in S3. It displays the spatial distribution of surface runoff of the Rouge River Watershed under S3 which had comparatively best performance in mitigating urban flooding among four scenarios.

Concerning the UFRM results for each of the four scenarios, none of the changes in urban flooding were particularly significant overall. However, we discovered that under

S3, the surface runoff in the site was comparatively smaller (Figure 5). In S3, there was a perceptible variation in surface runoff across the region. Generally, the south exhibited slightly higher surface runoff compared to the north. The areas with the most substantial surface runoff were concentrated in the site's southeastern corner and central location. Additionally, areas with significant runoff could also be found in the gently sloping terrain in the site's central and southern parts. Conversely, regions with relatively low surface runoff were mainly located in the central-southern section of the site, along the western boundary, and in a block-like pattern near the northern open boundary.

## 4. Discussion

### 4.1. Factors Influencing Historical Nutrient Export and Urban Flooding

#### 4.1.1. Factors Influencing Nutrient Export

The 2019 NDR model output revealed a significant correlation between N and P exports and their spatial association with the LULC categories, signifying a probable relationship between LULC and nutrient export. According to 2019's LULC map (Figure A2) and NDR results (Figure A4), the southeastern areas dominated by medium- to high-intensity urban LULC types were critical regions for increased N and P exports. This increase can be attributed to nutrient export from intensified urban intensity, industrial activities, expansion of impermeable surfaces, human activities, and fossil fuel consumption [48]. Indeed, several industrial facilities in the downstream Detroit area of the watershed could contribute to substantial point source pollution. It might also explain another phenomenon in the 2019 NDR output, which illustrated that the central-southern regions had fewer industrial facilities despite having many medium- to high-intensity developed areas. Consequently, the N and P exports were lower than in southeastern regions. Furthermore, other areas with severe N and P contamination—e.g., southern, central-southern, and some northwestern parts—were also dominated by medium- and high-intensity urban developed areas, likely due to similar reasons. Contrastingly, regions with lower nutrient exports were primarily located in the west, north, and some areas in the central and southeast, where multiple green spaces dominated the terrain. It can be attributed to the ecosystems comprising different plants, microorganisms, and animals, which effectively absorb N and P. Additionally, coupled with healthier soils that are better structured, more fertile, and have higher organic matter content, these green LULC types significantly contribute to environmental health. Their presence leads to more permeable surfaces that can intercept and retain a larger quantity of nutrients, thereby substantially reducing the input of N and P into aquatic systems like rivers and lakes [51,52].

We discovered significantly lower exports from P than N in our study area. This discrepancy was attributed mainly to the high urbanization levels within the watershed. Various relevant studies also demonstrate that urbanization—compared to promoting an increase in P—contributes more to N increase [53,54]. In addition, emissions from the extensive industrial facilities in the Rouge River Watershed and high concentrations of exhaust gases ($NH_3$) from dense traffic extensively elevate N pollution levels [55,56]. Furthermore, intensified N deposition due to climate change could also contribute to the disparity between N and P [57]. Another contributing factor could be that N output incorporates surface and subsurface export in the NDR model, while P output only considers surface exports. This is because particulate phosphorus—usually bound with sediments—is unlikely to migrate through subsurface flow. Consequently, NDR does not model subsurface phosphorus [6]. In addition to LULC, the differences between N and P could also be associated with the input biophysical table. One of the primary limitations of the NDR model is its high input sensitivity, signifying the need for further refinement in the biophysical table based on observed N and P in the watershed.

#### 4.1.2. Factors Influencing Urban Flooding

The 2019 UFRM results demonstrated hydrological dynamics aligned with LULC. To be specific, there is a conspicuous relationship between LULC and surface runoff. The

apparent runoff concentration was manifested in the southeast region, most conspicuously around the Detroit area, which was severely impacted by the urbanization threats in this area. Urbanization processes can significantly modify urban hydrological patterns and adversely affect both surface and downstream water bodies. This impact is primarily due to the proliferation of impervious surfaces, the eradication of deep-rooted vegetation, and modifications to the existing natural drainage systems [58–60]. Such changes often lead to reduced infiltration capacity and an increase in surface runoff [61–63]. Furthermore, these alterations have the potential to trigger significant flooding events [64–68]. Similarly, cities tend to encounter greater susceptibility to flooding during rainfall periods, with more runoff at the southwestern boundary of the watershed. Aside from the southeast corner of the site, the southwest direction also experiences high surface runoff. The primary reason might be the site's location in the downstream area of the Rouge River Watershed, which is characterized by an extensive network of rivers. During the rainy and flooding seasons, a substantial amount of water flows downstream from the upper reaches. Additionally, the hydrology soil group in this area is predominantly classified as C and D, known for their poor infiltration and drainage capabilities. Consequently, this region is more prone to higher surface runoff. In contrast, the lower surface runoff observed in the north can largely be credited to the surrounding wetland ecosystems. These wetlands serve as natural buffers, absorbing and moderating surface runoff. This capacity to store and slowly release water not only mitigates flood risks but also enhances the area's resilience to flooding. The wetlands' ability to trap sediments and nutrients further contributes to maintaining water quality, while their dense vegetation aids in soil stabilization and erosion control, reinforcing the overall integrity of the hydrological system in this region [69].

### 4.2. Factors Influencing Nutrient Export and Urban Flooding under Future Scenarios

4.2.1. Discussion of Future LULC Scenarios

Within the temporal frames of 2001, 2011, and 2019, our results demonstrated an increasing trend in medium- and high-intensity areas, conversely paralleled by the reduction of open spaces, a trend persisting into S1 (Figure A3). Furthermore, this scenario harmonizes with the historical LULC trends, exhibiting a significant decrease in forests, while forests can play an essential role in people's connection to nature and help mitigate the adverse effects of climate change by multiple functions like dissipating urban heat through evapotranspiration [70]. The reduction of forests may increase local climate change risk and increase the population's exposure to more considerable compounded risks [71].

The other three scenarios (S2, S3, and S4) attempt to improve the local environment and ecology (Figure A3). In the forest conservation scenario, we constrained the deciduous forest to transfer to open space and forced open space to transfer to the deciduous forest to protect the forest. The observed changes were also significant in this scenario. We noticed that the deciduous forest area increased under the forest protection scenario. In addition, the increasing area of low-intensity, medium-intensity, and high-intensity developed areas has been reduced compared with S1 due to our forced open space transferring to deciduous forests. However, this scenario still substantially reduces open space, and high-intensity urbanization continues.

Other scenarios, with their ecological emphasis, endeavor to modulate these impacts. For instance, S2 marks a shift toward environmental preservation by limiting the transformation of deciduous forests to open space. Explorations into S3 and S4 are associated with S2, introducing urbanization interventions. Among them, S3 evaluated the transition of deciduous forests to open spaces and forces transition from low- and medium-intensity to high-intensity developed regions. It necessitates the transformation of open spaces into deciduous forests, realizing a low-intensity development paradigm. This scenario has a reduced high-intensity developed area but an increase in medium-intensity developed areas of 13,151 acres (53.22 km$^2$). In contrast, S4 protects deciduous forests and open spaces, obstructing their conversion to more intensive developments and nurturing a low-intensity landscape (Figure 6). Moreover, open spaces experience an increase since it limits the

transformation from open space to low- and medium-intensity developed areas. It also optimizes the expanse of the forested regions and induces a tangible reduction in low-intensity developed areas, with a smaller increase in medium-intensity developed areas.

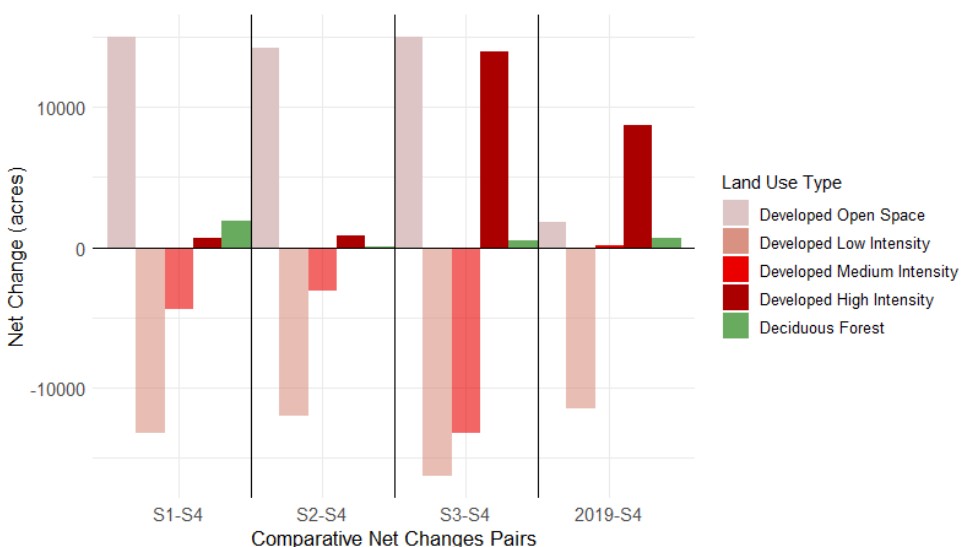

**Figure 6.** Bar chart of S4 net change in LULC with 2019 and S1, S2, S3 of Rouge River Watershed. From left to right, the figure illustrates the net changes in five land use/land cover (LULC) categories in the Rouge River Watershed for S4 compared to 2019 and S1, S2, and S3. These categories include open space, low-intensity, medium-intensity, high-intensity, and deciduous forest.

4.2.2. Factors Influencing Nutrient Export under Future Scenarios

S1 illustrated a significant change in urban LULC types, while the change in different green space LULC types remained similar. The increase in developed areas of varying intensities and the reduction of open space are very likely the main reasons for the relatively uniform growth of N and P. This is because urbanization can affect the cycles of N and P [72]. For urban watersheds, this primarily occurs through the reduction of N and P retention, thereby increasing their export [54,72,73]. Compared to other urban land uses, open spaces are more effective in intercepting nutrients, which will reduce the export of N and P [74,75]. Consequently, on a spatial scale, N and P are primarily concentrated in the northeastern part of the watershed could be attributed to the pronounced urbanization in the region. Since we utilized S1 as the baseline scenario, its numeric variations were referenced against the other three scenarios.

The slight increase in N and P export from S2 compared to S1 indicates that the forest conservation strategy adopted in S2 had minimal impact on N and P export. For LULC settings, S2 primarily restricted the transformation of deciduous forests into open spaces and enforced the reverse process, indicating the similar effectiveness of open spaces and deciduous forests in minimizing the export of N and P in this watershed. This is because both can reduce nutrient output through different mechanisms. For instance, deciduous forests can filter pollutants through a combination of soil absorption and plant uptake, while open spaces effectively reduce impervious surfaces, thereby diminishing the export of N and P [76–79]. According to NLCD, open space is predominantly covered by lawn grass and has less than 20% impervious surfaces [80]. Consequently, it will contribute to the effective reduction of N and P pollution to some extent. Although several studies suggest that deciduous forests have a more pronounced mitigating effect compared to open spaces, issues such as enhanced leaf litter leading to increased nutrient deposition may offset some of their benefits [81–83]. However, due to the relatively inconspicuous trends of two types of LULC transformations under S2 during the watershed's history and the neglect of other subtle green space LULC transformation types in this research study, the ecological benefits of S2—based on forest protection in the Rouge River Watershed—may not be pronounced.

S3 indicated a surge in N and P levels compared to S1 and S2 specifying the mitigating influence of low-intensity development on N and P exports. The spatial changes under S3 (Figure A4) may correlate with the conversion of open spaces to medium-intensity developed areas in the central region and the reduction of high-intensity developed areas in the northeast. These phenomena likely indicate that elevated N and P export is sensitive to open space development, while decreased export is sensitive to the degradation of high-intensity developed regions. The urban expansion due to the transition from open space to medium intensity will result in a significant reduction of impervious surfaces and a decrease in vegetation coverage [81,84,85]. This will in turn lead to an increase in TP, TN, and total suspended solids (TSS) in the watershed [85–88]. Conversely, the degradation of urban land can alter the nutrient cycling processes in the original soil layers, potentially reducing the output of N and P to some extent [89–91]. Therefore, preserving open space and reducing high-intensity developed areas may positively affect minimizing N and P output. However, the efficacy of S3 in reducing N and P levels was accomplished by reducing levels in most areas while increasing them in several others, hinting at conceivable resource wastages because of preventable land degradation and development. Furthermore, the reduction of high-intensity developed areas can also increase the economic and policy burdens of the area [92]. Combined with regional historical conditions in Detroit, this transformation is likely to result in a significant number of vacant lots [93]. Hence, while S3 effectively lowered N and P levels, its applicability—from the perspective of urban development—requires further validation and investigation.

Finally, S4 demonstrated the most significant improvement in N and P export levels among all scenarios compared to S1. The variation in S4 lay in increased open spaces and reduced low- and medium-intensity developed areas. The latter reduction mainly originated from the restrictions on developing open spaces, showing the efficacy of conserving open spaces in reducing N and P pollution. In fact, open spaces play a crucial role in reducing N and P pollution through the combination of natural processes and ecological functions [94,95]. These green spaces, composed of trees, grasslands, and wetlands are capable of improving water quality, reducing air pollution, and minimizing soil erosion [96–98]. With more reasonable planning and design in the future, these open spaces can be transformed into various green infrastructures to further enhance their ecological functions [99–101]. Additionally, these open spaces can serve as corridors, nodes, and patches in ecological structures at a broader scale and contribute to forming a regional ecological system in the future [102,103]. Compared to other scenarios spatially, LULC transitions in S4 were comparatively more uniform, favoring future human interventions and landscape restoration measures in localized areas. However, possibly due to the reduction in low- and medium-intensity developed regions, we could not observe substantial growth in high-intensity developed areas in S4. Hence, while S4 performed well ecologically, it might have potentially harmful implications for urban construction and economic development.

### 4.2.3. Factors Influencing Urban Flooding under Future Scenarios

Under the S3 scenario, although the conversion between land covers of varying development intensities leads to an increase in the developed area in the central part of the site, thereby increasing surface runoff in some areas of the central and southern parts, the area of increased runoff is larger than in other scenarios. However, regarding net change, the extent of increased surface runoff in these areas is not significant. Conversely, in the northern part of the site, due to the transformation of medium- and high-intensity development areas into lower-intensity development areas and open spaces, there is a notably significant reduction in surface runoff. This indicates that the urban flooding transition under S3 satisfies the sustainable development goals for the area well [104].

The differences in outcomes between the four scenarios (S1–S4) remained minimal when evaluated quantitatively, which reveals that changes in urban impervious areas and LULC do have a localized impact on surface runoff variability. Therefore, the overall

volume remains reasonably constant. This finding is convincing since from a hydrological perspective, LULC change within a watershed has been recognized as one of the critical factors influencing runoff generation, and the change in runoff was highly correlated with the percentage of highly developed areas [105–107]. For instance, in the case of S1 and S2, while S1 illustrated a more pronounced surge in surface runoff in the site's northern boundary than S2, the latter compensated with a more extensive runoff increase in its central and southern regions.

*4.3. Planning Implications*

Safeguarding open spaces and limiting urbanization in the originally developed areas of the Rouge River Watershed were crucial (same as S4). Therefore, this requirement necessitates strategic utilization of existing infrastructure to minimize sprawl and its ecological repercussions. Accomplishing this objective involves consolidating functionalities, mitigating commutes, lessening congestion, and optimizing energy utilization. Particularly in urban centers like Detroit, it is vital to promote a compact, pedestrian-friendly ecosystem that mitigates overreliance on vehicles and urban transportation. Simultaneously, the inception of green corridors is critical, reinforcing urban biodiversity, improving wildlife mobility, and offering recreational avenues for citizens. To achieve this goal, integrating green infrastructure—e.g., green roofs and rain gardens—could modulate urban microclimates and create venues for urban agriculture, aligning with community-driven strategies.

Furthermore, a green infrastructure plan—bridging the gap between preserved and developed spaces—stands out as an optimal preservation strategy. It attenuates nutrient export and flood damage by controlling stormwater runoff, moderating peak flow rates, and ameliorating water quality [108].

Strategically conserving open space is vital for the Rouge River watershed, offering a long-term solution to mitigate nutrient exports aligning with sustainable urban planning principles. Several strategies can optimize open space preservation: (1) Collaborating with stakeholders to specify and secure vital open areas; (2) Advocating for policies that motivate private conservation initiatives; (3) Empowering communities to enhance open spaces for the creation of green systems is a crucial strategy that underscores inclusive community involvement. This method involves engaging in open-ended, personal dialogues before any development starts and tackles regional stormwater issues, while simultaneously preserving each community's unique character and meeting the aesthetic standards of their specific design needs [109–111]. Additionally, it ensures the placement of essential stormwater management facilities where they are most needed. Moreover, converting available land into community gardens or open spaces filled with native plants and trees enhances the area's capacity to absorb rainwater [112–114]. This practice not only aids in flood prevention but also contributes to the ecological and aesthetic value of the community. Moreover, the adoption of smart water management technologies, such as sensors for monitoring rainfall and water levels, offers an advanced approach to early warning systems and rapid response mechanisms; (4) Enhancing public participation in open spaces and popularizing eco-education. By preserving open spaces, urban regions can better enhance their ecological resilience and eliminate the adverse impacts of nutrient export and urban flooding. Additionally, the protection and intelligent design of open spaces can effectively enhance resilience to natural disasters during the process of urban development [115]. These open areas can serve as several types of green infrastructure, providing functions such as stormwater management, urban shoreline protection, and emergency refuge during natural disasters [116,117]. For instance, the green open spaces in Jamaica Bay, New York City, have mitigated the impact of hurricanes and storms many times, ensuring the safety of residents [118]. In economic terms, protecting open spaces—especially the application of green infrastructure—can enrich the area's functionality, bringing in tax revenues. Furthermore, the appeal of urban green spaces can generate co-benefits in improving property values and encouraging tourism, further promoting local economic

development [119,120]. Lastly, the design of open spaces must also take into consideration factors such as visual elements, as these factors directly influence people's subjective perception of the site [121].

*4.4. Limitation*

Both urban development and environmental protection management measures are important considerations in scenario design. For our study, we did not include them as we intended to explore the efficiency of different LULC strategies for mitigating nutrient export and flood risk at the planning scale. Therefore, one limitation of this research study is that all four alternative scenarios (S1–S4) were somewhat extreme in their predictions. It is because we aimed to discern the degrees of nutrient export and urban flooding more distinctly due to various LULC transitions. Therefore, we retained only the most pronounced types of urban land transitions. However, the actual process of urban development could be more intricate. Additionally, given that the Rouge River Watershed is highly urbanized and various green land cover types comprise only a small proportion, transitions related to green land types have been very scarce, except for the conversion from deciduous forest to open space. Consequently, in simulating future urban development scenarios, we have omitted the majority of transitions associated with green land types, which may render the ecological advantages of S2 insignificant. For future enhancements, LULC transitions associated with green land types could be incorporated into the LCM as variables for consideration. During the model's algorithmic learning process, more encompassing driving variables like the distance between various LULCs, roads, and populations could be integrated to improve LCM's simulation accuracy. Furthermore, the NDR and UFRM models performed comparatively better, but both could be further enhanced. Both NDR and UFRM models are commonly used for their simplicity and robustness. Due to the difficulty in obtaining measured runoff depth/volume data in the UFRM model and nutrient retention data in the NDR model, past studies often do not perform model calibration [6,9,122,123]. Although we have conducted calibration based on the correlation between nutrient export and the SQI dataset, there is still room for improvement in terms of observed data in the future [124–128]. However, in general, the results of the model calibration meet the research needs. Given the relative simplicity of their mechanisms, the results produced by these models exhibited high sensitivity to the input files (e.g., the biophysical table). Thus, model inputs could still be enhanced, and some parameters—such as the TFA and kb in NDR and rainfall depth in UFRM—could be calibrated and modified using empirical data.

**5. Conclusions**

Among the four scenarios, S4—aiming to restrain the development of open spaces—demonstrated the most effective nutrient export reduction of the Rouge River Watershed, in which N and P were reduced by 2.94% and 2.58%, respectively, compared to the baseline scenario. The mitigation effects of urban flooding do not differ substantially across the four scenarios (S1–S4). Therefore, we conclude that S4 may produce the most significant ecological benefits for the watershed. It implies that the aquatic environmental issues in this watershed are quite sensitive to developing open spaces. Thus, preserving open spaces should be a principal focus for this watershed's future sustainable development projects. However, as S4 may influence economic development within the watershed, we also propose appropriate green infrastructure plans to minimize the associated economic tradeoffs while preserving open spaces. Among the other three scenarios (S1–S3), S1 and S2 did not demonstrate notable improvements in nutrient export. S1 increased N and P by 8.37% and 8.86%, respectively, compared to 2019, while S2 increased by 8.47% and 9.42%, highlighting that the ecological benefits of only preserving forests may not be practical in this watershed. While S3 could effectively decrease nutrient export, in which N and P were reduced by 2.79% and 1.10%, respectively, compared to S1, its disadvantages were conspicuous primarily due to its larger-scale modifications to the original LULC. However, the four scenarios' simulations also have some limitations as we have focused only on the

nine primary LULC transitions within the watershed. Although the transformations in multiple other LULC categories did not manifest prominently in the watershed, including them could still improve the accuracy of the predicted LULC maps.

Our proposed model also validates the feasibility of using InVEST in conjunction with LCM to predict the correlations between LULC and urban flooding and nutrient export. Although NDR and UFRM models have some inherent limitations, they are valuable for macro-level water ecological analysis and quantification. Our methodology can be applied to future studies at similar watershed scales with similar urbanization levels and other climate and land cover characteristics. Due to the simplicity and robustness of the InVEST and LCM models, this method workflow can efficiently analyze and predict nutrient outputs and flood risks of watersheds in general. However, NLCD maps are only available for study areas in the U.S. Study areas in other counties require similar resolution of land cover maps (30 m). Additionally, integrating specific natural and socioeconomic conditions (e.g., slope, road network) of the site can effectively improve the prediction accuracy of the LCM, which is suitable for other study sites with corresponding data layers. From the revaluation results of the four scenarios (S1–S4), we can determine their difference and specify more sustainable urban development scenarios. However, the performances of both models can be enhanced by using more accurate model inputs and parameter configurations in the future.

**Author Contributions:** Conceptualization, Y.Z. and R.W.; methodology, Y.Z., Y.R. and R.W.; software, Y.Z. and Y.R.; validation, Y.Z. and Y.R.; formal analysis, Y.Z., Y.R. and T.L.; investigation, Y.Z. and Y.R.; resources, R.W.; data curation, Y.Z. and Y.R.; writing—original draft preparation, Y.Z., Y.R., Y.L. and T.L.; writing—review and editing, R.W.; visualization, Y.L., T.L., L.K. and Q.D.; project administration, Y.Z. All authors have read and agreed to the published version of the manuscript.

**Funding:** This research received no external funding.

**Data Availability Statement:** Data are contained within the article.

**Conflicts of Interest:** The authors declare no conflict of interest.

**Appendix A**

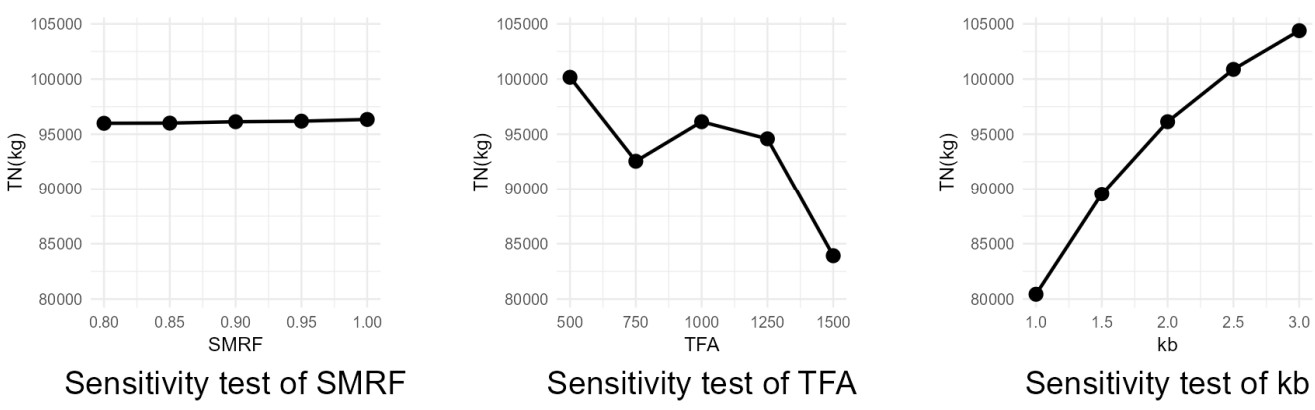

**Figure A1.** Diagrams of sensitivity test of NDR parameters. The figure sequentially displays the results of sensitivity analyses for parameters SMRF, TFA, and kb where the default values of the three parameters are 0.9, 1000, and 2. We use TN (kg) as a basis here since SMRF is only related to N, not P.

**Table A1.** The correlation of NDR results of different combinations of TFA and kb with SQI.

| TFA | kb | Correlation P [1] | Correlation N [2] |
|---|---|---|---|
| 2000 | 8 | 0.494298 | 0.57489 |
| 2000 | 4 | 0.494298 | 0.57489 |
| 2000 | 2 | 0.544141 | 0.612917 |
| 2000 | 1 | 0.599834 | 0.667834 |
| 2000 | 0.5 | 0.581968 | 0.642988 |
| 1000 | 8 | 0.567762 | 0.604707 |
| 1000 | 4 | 0.567763 | 0.605331 |
| 1000 | 2 | 0.571693 | 0.62014 |
| 1000 | 1 | 0.629356 [3] | 0.677009 [3] |
| 1000 | 0.5 | 0.582942 | 0.655857 |
| 800 | 8 | 0.133197 | 0.135486 |
| 800 | 4 | 0.133222 | 0.135511 |
| 800 | 2 | 0.133201 | 0.135499 |
| 800 | 1 | 0.133193 | 0.135502 |
| 800 | 0.5 | 0.133227 | 0.135542 |
| 500 | 8 | 0.133197 | 0.135486 |
| 500 | 4 | 0.133177 | 0.135464 |
| 500 | 2 | 0.133165 | 0.135454 |
| 500 | 1 | 0.135472 | 0.133163 |
| 500 | 0.5 | 0.133171 | 0.135474 |
| 200 | 8 | 0.041551 | 0.04784 |
| 200 | 4 | 0.041553 | 0.047854 |
| 200 | 2 | 0.041543 | 0.047845 |
| 200 | 1 | 0.047843 | 0.04154 |
| 200 | 0.5 | 0.041552 | 0.047862 |

[1] Correlation of P output of NDR with SQI; [2] Correlation of N output of NDR with SQI; [3] The highest correlation when TFA = 1000, kb = 1.

**Table A2.** Nine most significant LULC changes in the Rouge River Watershed from 2001 to 2019.

| No. | Previous LULC Type | Following LULC Type |
|---|---|---|
| 1 | Developed, open space | Developed, low intensity [1] |
| 2 | Developed, open space | Developed, medium intensity [2] |
| 3 | Developed, low intensity | Developed, open space |
| 4 | Developed, low intensity | Developed, medium intensity |
| 5 | Developed, low intensity | Developed, high intensity [3] |
| 6 | Developed, medium intensity | Developed, low intensity |
| 7 | Developed, medium intensity | Developed, high intensity |
| 8 | Developed, high intensity | Developed, medium intensity |
| 9 | Deciduous forest | Developed, medium intensity |

[1] Areas with a mixture of constructed materials and vegetation. Impervious surfaces account for 20% to 49% percent of total cover; [2] Areas with a mixture of constructed materials and vegetation. Impervious surfaces account for 50% to 79% of the total cover; [3] Highly developed areas where people reside or work in high numbers. Impervious surfaces account for 80% to 100% of the total cover.

**Appendix B**

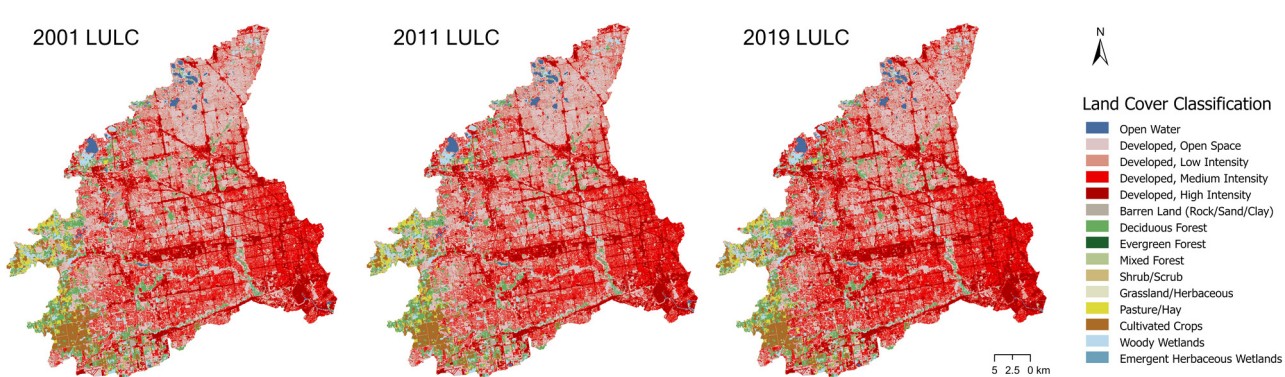

**Figure A2.** Map of historical LULC of Rouge River Watershed. The figure sequentially displays the land use/land cover (LULC) of the Rouge River Watershed for the years 2001 (**left**), 2011 (**middle**), and 2019 (**right**).

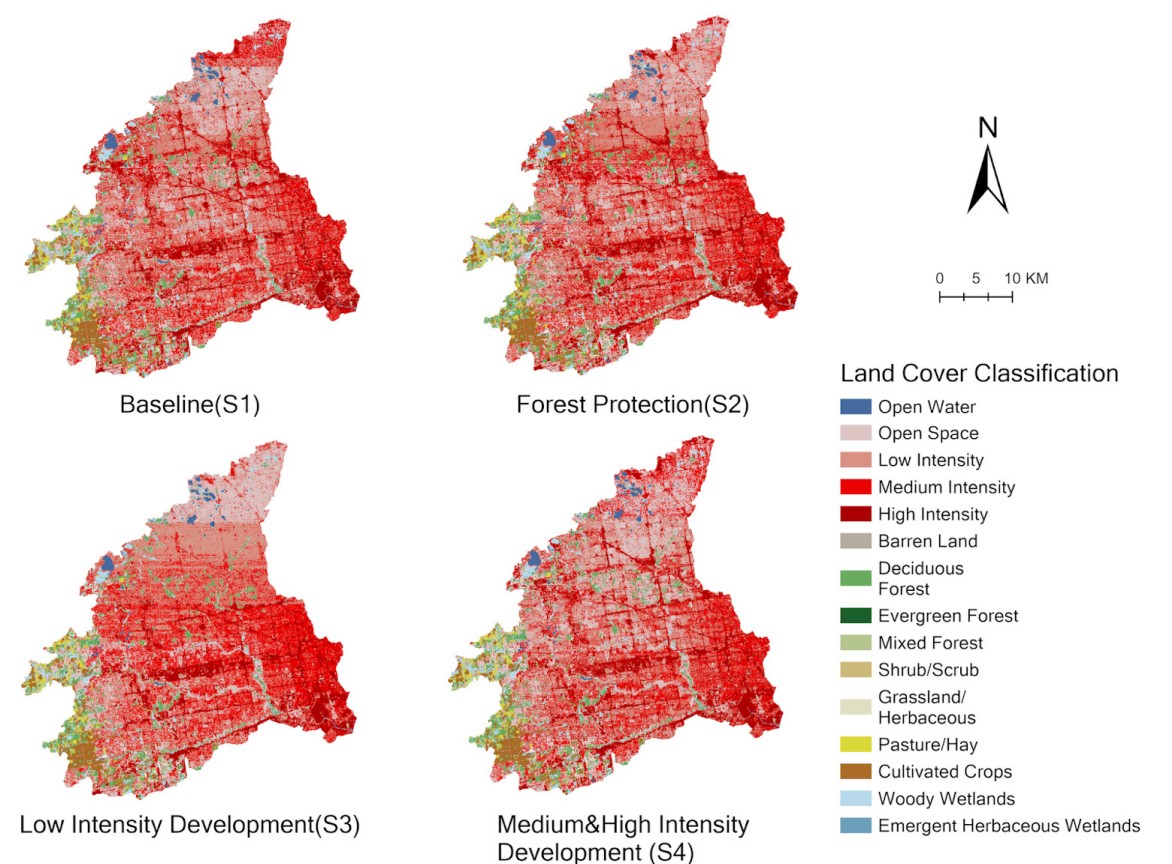

**Figure A3.** Map of future LULC of Rouge River Watershed. The figure sequentially displays the land use/land cover (LULC) of the Rouge River Watershed for four scenarios: S1 (**upper left**), S2 (**upper right**), S3 (**lower left**), and S2 (**lower right**).

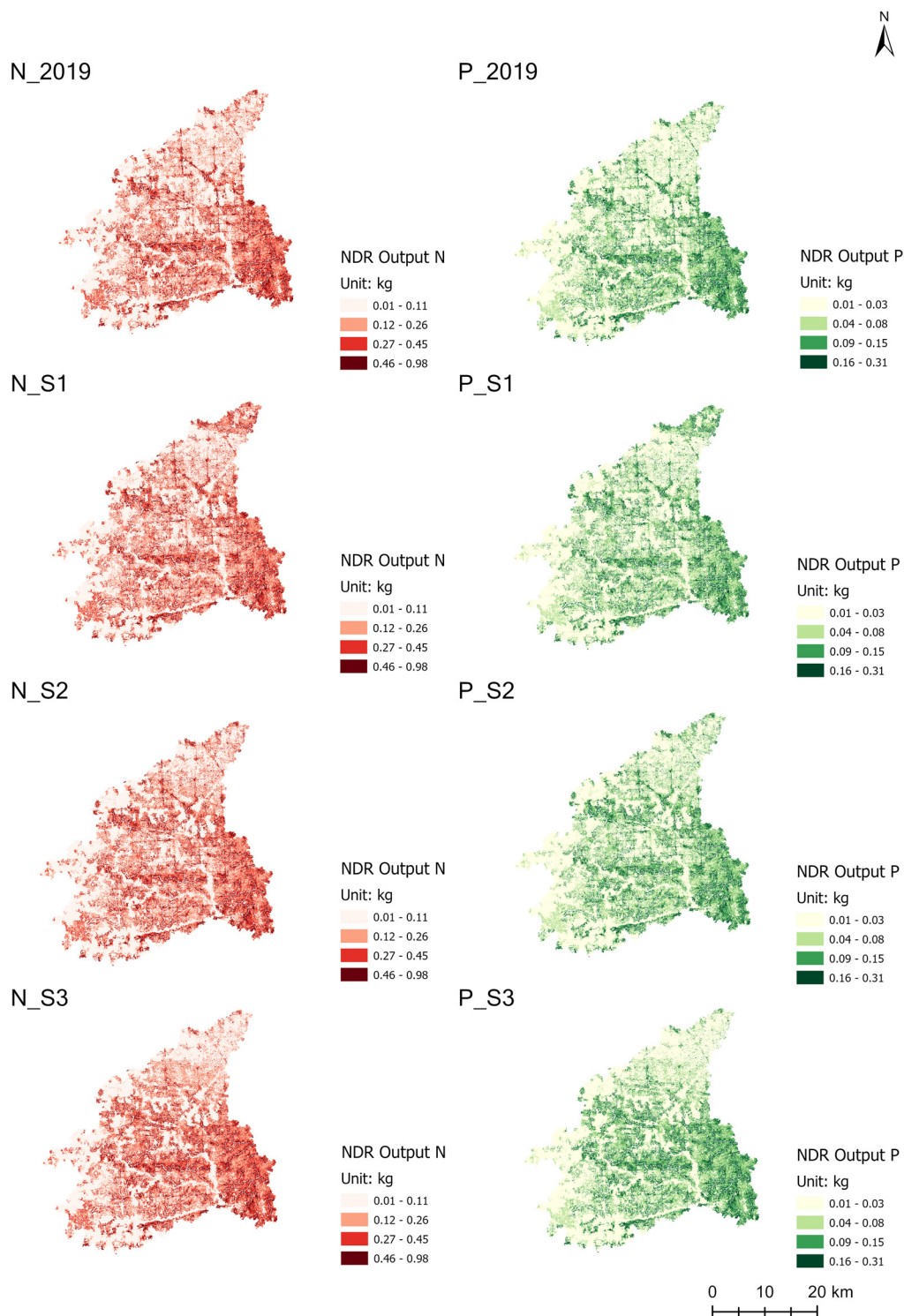

**Figure A4.** Map of the NDR results for the Rouge River Watershed in 2019 and S1, S2, and S3. The figure sequentially displays the results for N and P in the Rouge River Watershed for 2019 and the four scenarios (from top to bottom: 2019, S1, S2, S3). Red represents N, and green represents P.

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
