# Peer review of "Investigating Urban Flooding and Nutrient Export under Different Urban Development Scenarios in the Rouge River Watershed in Michigan, USA"

_land, doi:10.3390/land12122163_

Round 1
Reviewer 1 Report
Comments and Suggestions for Authors
In this study, the authors simulated the changes of land use, nitrogen and phosphorus output load and surface runoff in the Rouge River Watershed under different urban development scenarios. Some integrative planning and landscape strategies were proposed.
Before the article is accepted, I would like the authors to answer the following questions and try to make major changes to the manuscript.
(1)The authors should state the results of the model calibration, including LCM model、NDR model and UFRM model, how accurate is it? Does it meet the research needs? Please show a comparison between the observations and the simulated results.
(2)Please provide additional information on the setting of the grid cell size in the study area and the driving factors considered in the LCM model.
(3) The title and content of Part 4.1 do not seem to match. I am not sure what is being evaluated. It feels more like an analysis of the factors that might influence the spatio-temporal pattern of nutrient export and urban flooding.
(4)What does "After Revaluation" in the title 4.2 mean? This part includes the comparative analysis of the simulation results of different scenarios, and it is better to state it in the result section.
(5)It is recommended to further analyze the factors affecting nutrient output load intensity and its spatial differences in the discussion section, as well as the analysis of model sensitivity and rationality results.
(6)The LULC scenarios are designed by restricting the types of land use change. It is relatively simple. It is recommended to consider urban development and environmental protection management measures, such as increasing green infrastructure and optimizing the landscape pattern to improve ground permeability and strengthen flood retention.
Reviewer 2 Report
Comments and Suggestions for Authors
Reviewer 3 Report
Comments and Suggestions for Authors
The article is an interesting study that uses different models to show the consequences of urban flooding in the urban development process.
The article shows the comparison that results from the application of different models in order to achieve optimal scenarios for urban development. It is correctly written and may be useful for those involved in the analysis of issues affecting the determinants of riverine city development.
Comments on the Quality of English LanguageThe language of the article is clear and requires little correction at most stylistically
Author Response
We appreciate your comments and suggestions. We have further refined the content of the article, mainly including model sensitivity analysis, more informed discussion, and writing improvement.
Reviewer 4 Report
Comments and Suggestions for Authors
Dear Authors,
I read the article with interest The topic under discussion addressing the specification of LULC types that can effectively mitigate the Rouge River Watershed's flooding and nutrient export issues for sustainable urban development. In my opinion your manuscript suits the profile of the journal Land. However, I have some comments which should be taken into consideration before publishing:
1. One of the research objective was to quantify historical urban flooding and nutrient export of the Rouge River Watershed. In my opinion, this task has not been accomplished. Please complete the missing content.
2. Please always use two units of measurements (e.g. square meters and miles) at the same time.
3. Fig. 1. The LULC map should be shown separately with a clear/readable legend.
4. Please provide information about the scale (accuracy) of maps LULC and Runoff potential map.
5. The Discussion section largely resembles the description of the results. On two pages (p.12-13) there is no reference to other articles at all. Please move the descriptive part to the Results section and focus on the discussion in the Discussion section.
6. Please indicate to what extent the methodology used can be transferred to other geographical conditions. What is the size of the catchment area, what is the accuracy of the data, maps...
7. Please describe the river watershed in more detail, e.g. detailed quantitative and qualitative characteristics of land use, share of agriculture (including urban gardening), which is important in providing nutrients.
Best regards!
Comments on the Quality of English LanguageMinor editing of English language required.
